# Neural Linear Bandits: Overcoming Catastrophic Forgetting through Likelihood Matching

## Abstract

We study neural-linear bandits for solving problems where *both* exploration and representation learning play an important role. Neural-linear bandits leverage the representation power of deep neural networks and combine it with efficient exploration mechanisms, designed for linear contextual bandits, on top of the last hidden layer. Since the representation is being optimized during learning, information regarding exploration with "old" features is lost. Here, we propose the first limited memory neural-linear bandit that is resilient to this catastrophic forgetting phenomenon. We perform simulations on a variety of real-world problems, including regression, classification, and sentiment analysis, and observe that our algorithm achieves superior performance and shows resilience to catastrophic forgetting.

## 1 Introduction

Deep neural networks (DNNs) can learn representations of data with multiple levels of abstraction and have dramatically improved the state-of-the-art in speech recognition, visual object recognition, object detection and many other domains such as drug discovery and genomics (LeCun et al., 2015; Goodfellow et al., 2016). Using DNNs for function approximation in reinforcement learning (RL) enables the agent to generalize across states without domain-specific knowledge, and learn rich domain representations from raw, high-dimensional inputs (Mnih et al., 2015; Silver et al., 2016).

Nevertheless, the question of how to perform efficient exploration during the representation learning phase is still an open problem. The $\epsilon$-greedy policy (Langford & Zhang, 2008) is simple to implement and widely used in practice (Mnih et al., 2015). However, it is statistically suboptimal. Optimism in the Face of Uncertainty (Abbasi-Yadkori et al., 2011; Auer, 2002, OFU), and Thompson Sampling (Thompson, 1933; Agrawal & Goyal, 2013, TS) use confidence sets to balance exploitation and exploration. For DNNs, such confidence sets may not be accurate enough to allow efficient exploration. For example, using dropout as a posterior approximation for exploration does not concentrate with observed data (Osband et al., 2018) and was shown empirically to be insufficient (Riquelme et al., 2018). Alternatively, pseudo-counts, a generalization of the number of visits, were used as an exploration bonus (Bellemare et al., 2016; Pathak et al., 2017). Inspired by tabular RL, these ideas ignore the uncertainty in the value function approximation in each context. As a result, they may lead to inefficient confidence sets (Osband et al., 2018).

Linear models, on the other hand, are considered more stable and provide accurate uncertainty estimates but require substantial feature engineering to achieve good results. Additionally, they are known to work in practice only with "medium-sized" inputs (with around $1,000$ features) due to numerical issues. A natural attempt at getting the best of both worlds is to learn a linear exploration policy on top of the last hidden layer of a DNN, which we term the **neural-linear** approach. In RL, this approach was shown to refine the performance of DQNs (Levine et al., 2017) and improve exploration when combined with TS (Azizzadenesheli et al., 2018) and OFU (O'Donoghue et al., 2018; Zahavy et al., 2018a). For contextual bandits, Riquelme et al. (2018) showed that neural-linear TS achieves superior performance on multiple data sets.

A practical challenge for neural-linear bandits is that the representation (the activations of the last hidden layer) change after every optimization step, while the features are assumed to be fixed over time when used by linear contextual bandits. Riquelme et al. (2018) tackled this problem by storing the entire data set in a memory buffer and computing new features for all the data after each DNN learning phase. The authors also experimented with a bounded memory buffer, but observed a significant decrease in performance due to **catastrophic forgetting** (Kirkpatrick et al., 2017), i.e., a loss of information from previous experience.

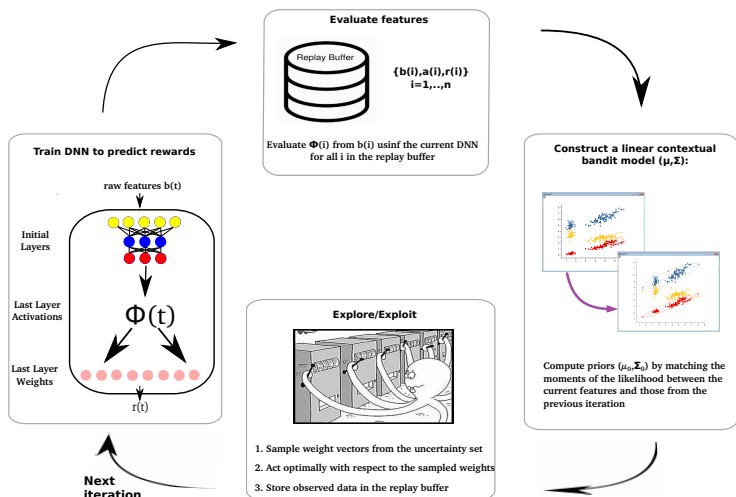

Figure 1: Neural-Linear contextual Thompson sampling with limited memory.

In this work, we propose a neural-linear bandit that uses TS on top of the last layer of a DNN (Fig. 1)[1]. Key to our approach is a novel method to compute priors whenever the DNN features change that makes our algorithm resilient to catastrophic forgetting. Specifically, we adjust the moments of the likelihood of the reward estimation conditioned on new features to match the likelihood conditioned on old features. We achieve this by solving a semi-definite program (Vandenberghe & Boyd, 1996, SDP) to approximate the covariance and using the weights of the last layer as prior to the mean.

We present simulation results on several real-world and simulated data sets, including classification and regression, using Multi-Layered Perceptrons (MLPs). Our findings suggest that using our method to approximate priors improves performance when memory is limited. Finally, we demonstrate that our neural-linear bandit performs well in a sentiment analysis data set where the input is given in natural language (of size $\mathbb{R}^{8k}$) and we use a Convolution Neural Network (CNNs). In this regime, it is not feasible to use a linear method due to computational problems. To the best of our knowledge, this is the first neural-linear algorithm that is resilient to catastrophic forgetting due to limited memory.

## 2 BACKGROUND

**The stochastic, contextual (linear) multi-armed bandit problem.** There are $N$ arms (actions). At time $t$ a context vector $b(t) \in \mathbb{R}^d$, is revealed. The history at time $t$ is defined to be $H_{t-1} = \{b(\tau), a(\tau), r_{a(\tau)}(\tau), \tau = 1, ..., t-1\}$, where $a(\tau)$ denotes the arm played at time $\tau$. The contexts $b(t)$ are assumed to be **realizable**, i.e., the reward for arm $i$ at time $t$ is generated from an (unknown) distribution s.t. $\mathbb{E}[r_i(t)|b(t), H_{t-1}] = \mathbb{E}[r_i(t)|b(t)] = b(t)^T \mu_i$, where $\{\mu_i \in \mathbb{R}^d\}_{i=1}^N$ are fixed but unknown parameters. An algorithm for this problem needs to choose at every time $t$ an arm $a(t)$ to play, with the knowledge of history $H_{t-1}$ and current context $b(t)$. Let $a^*(t)$ denote the optimal arm at time t, i.e. $a^*(t) = \arg \max_i b(t)^T \mu_i$, and let $\Delta_i(t)$ the difference between the mean rewards of the optimal arm and of arm $i$ at time $t$, i.e., $\Delta_i(t) = b(t)^T \mu_{a^*(t)} - b(t)^T \mu_i$. The objective is to minimize the total regret $R(T) = \sum_{t=1}^T \Delta_{a(t)}$, where the time horizon T is finite.

---

**Algorithm 1** TS for linear contextual bandits

$\forall i \in [1.., N]$, set $B_i = I_d$, $\hat{\mu}_i = 0_d$, $f_i = 0_d$
**for** $t = 1, 2, \ldots,$ **do**
  $\forall i \in [1.., N]$, sample $\tilde{\mu}_i$ from $N(\hat{\mu}_i, v^2 B_i^{-1})$
  Play arm $a(t) := \arg \max_i b(t)^T \tilde{\mu}_i$
  Observe reward $r_t$
  **Update:** $B_{a(t)} = B_{a(t)} + b(t)b(t)^T$
  $f_{a(t)} = f_{a(t)} + b(t)r_t$, $\hat{\mu}_{a(t)} = B_{a(t)}^{-1} f_{a(t)}$
**end for**

---

**TS for linear contextual bandits.** Thompson sampling is an algorithm for online decision problems where actions are taken sequentially in a manner that must balance between exploiting what is known to maximize immediate performance and investing to accumulate new information that may improve future performance (Russo et al., 2018; Lattimore & Szepesvári, 2018). For linear contextual bandits, TS was introduced in (Agrawal & Goyal, 2013, Alg. 1).

---

[1]Image credits: bandit (bottom), Microsoft research; confidence ellipsoid (right), OriginLab.

Suppose that the **likelihood** of reward $r_i(t)$, given context $b(t)$ and parameter $\mu_i$, were given by the pdf of Gaussian distribution $N(b(t)^T\mu_i, \nu^2)$, and let $B_i(t) = B_i^0 + \sum_{\tau=1}^{t-1} b(\tau)b(\tau)^T \mathbb{1}_{i=a(\tau)}$, $\hat{\mu}_i(t) = B_i^{-1}(t)\sum_{\tau=1}^{t-1} b(\tau)r_{a(\tau)}(\tau)\mathbb{1}_{i=a(\tau)}$, where $\mathbb{1}$ is the indicator function. Given a Gaussian **prior** for arm $i$ at time $t$, $N(\hat{\mu}_i(t), v^2 B_i^{-1}(t))$, the **posterior** distribution at time $t+1$ is given by,

$$Pr(\tilde{\mu}_i|r_i(t)) \sim Pr(r_i(t)|\tilde{\mu}_i)Pr(\tilde{\mu}_i) \sim N(\hat{\mu}_i(t+1), v^2 B_i^{-1}(t+1)). \tag{1}$$

At each time step $t$, the algorithm generates samples $\{\tilde{\mu}_i(t)\}_{i=1}^N$ from the posterior distribution $N(\hat{\mu}_i(t), v^2 B_i^{-1}(t))$, plays the arm $i$ that maximizes $b(t)^T\mu_i(t)$ and updates the posterior. TS is guaranteed to have a total regret at time $T$ that is not larger than $O(d^{3/2}\sqrt{T})$, which is within a factor of $\sqrt{d}$ of the information-theoretic lower bound for this problem. It is also known to achieve excellent empirical results (Lattimore & Szepesvári, 2018).

Although that TS is a Bayesian approach, the description of the algorithm and its analysis are prior-free, i.e., the regret bounds will hold irrespective of whether or not the actual reward distribution matches the Gaussian likelihood function used to derive this method (Agrawal & Goyal, 2013).

**Bayesian Linear Regression.** A different mechanism, based on Bayesian Linear Regression, was proposed by Riquelme et al. (2018). Here, the noise parameter $\nu$ (Alg. 1) is replaced with a prior belief that is being updated over time. The **prior** for arm $i$ at time $t$ is given by $Pr(\tilde{\mu}_i, \tilde{\nu}_i^2) = Pr(\tilde{\nu}_i^2)Pr(\tilde{\mu}_i|\tilde{\nu}_i^2)$, where $Pr(\tilde{\nu}_i^2)$ is an inverse-gamma distribution Inv-Gamma$(a_i(t), b_i(t))$, and the conditional prior density $Pr(\tilde{\mu}_i|\tilde{\nu}_i^2)$ is a normal distribution, $Pr(\tilde{\mu}_i|\tilde{\nu}_i^2) \propto \mathcal{N}\left(\hat{\mu}_i(t), \tilde{\nu}_i^2 B_i(t)^{-1}\right)$. For Gaussian **likelihood**, the **posterior** distribution at time $\tau = t+1$ is, $\Pr(\tilde{\nu}_i) = $ Inv-Gamma$\left(a_i(\tau), b_i(\tau)\right)$ and $\Pr(\tilde{\mu}_i|\tilde{\nu}_i) = \mathcal{N}\left(\hat{\mu}_i(\tau), \nu_i^2 B_i(\tau)^{-1}\right)$, where:

$$B_i(t) = B_i^0 + \sum_{\tau=1}^{t-1} b(\tau)b(\tau)^T \mathbb{1}_{i=a(\tau)}, \;\; f_i(t) = \sum_{\tau=1}^{t-1} b(\tau)r_i(\tau)\mathbb{1}_{i=a(\tau)},$$

$$\hat{\mu}_i(t) = B_i(t)^{-1}\left(B_i^0\mu_i^0 + f_i(t)\right), \;\; a_i(t) = a_i^0 + \frac{t}{2}, \;\;\; R_i^2(t) = R_i^2(t-1) + r_i^2$$

$$b_i(t) = b_i^0 + \frac{1}{2}\left(R_i^2(t) + (\mu_i^0)^T B_i^0 \mu_i^0 - \hat{\mu}_i(t)^T B_i(t)\hat{\mu}_i(t)\right). \tag{2}$$

The problem with this approach is that the marginal distribution of $\mu_i$ is heavy tailed (multi-variate t-student distribution, see O'Hagan & Forster (2004), page 246, for derivation), and does not satisfy the necessary concentration bounds for exploration in (Agrawal & Goyal, 2013; Abeille et al., 2017). Thus, in order to analyze the regret of this approach, new analysis has to be derived, which we leave to future work. Empirically, this update scheme was shown to convergence to the true posterior and demonstrated excellent empirical performance (Riquelme et al., 2018). This can be explained by the fact that the mean of the noise parameter $\nu_i$, given by $\mathbb{E}\nu_i = \frac{b_i(t)}{a_i(t)-1}$, is decreasing to zero with time, which may compensate for the lack of shrinkage due to the heavy tail distribution.

## 3 LIMITED MEMORY NEURAL-LINEAR TS

Our algorithm, as depicted in Fig. 1, is composed of four main components: (1) A DNN that takes a raw context as an input and is trained to predict the reward of each arm; (2) An exploration mechanism that uses the last layer activations of the DNN as features and performs linear TS on top of them; (3) A memory buffer that stores previous experience; (4) A likelihood matching mechanism that uses the memory buffer and the DNN to account for changes in representation. We now explain how each of these components works; **code** can be found in (link).

To derive our algorithm we make the **assumption** that all the representations that are produced by the DNN are **realizable**. That is, for each representation there exist a *different* linear coefficients vector (e.g. $\mu$ for $\phi$, $\beta$ for $\psi$,) such that the expected reward is linear in the features. Explicitly, this means that for representations $\phi, \psi$ it holds that $\mathbb{E}[r_i(t)|\phi(t)] = \phi(t)^T\mu_i = \psi(t)^T\beta_i = \mathbb{E}[r_i(t)|\psi(t)]$.

While the realizability assumption is standard in the existing literature on contextual multi-armed bandits (Chu et al., 2011; Abbasi-Yadkori et al., 2011; Agrawal & Goyal, 2013), it is quite strong and may not be realistic in practice. We further discuss these assumptions in the discussion paragraph below and in Section 5.

**1. Representation.** Our algorithm uses a DNN, denoted by $D$, that takes the raw context $b(t) \in \mathbb{R}^d$ as its input. The network has $N$ outputs that correspond to the estimation of the reward of each arm; given context $b(t) \in \mathbb{R}^d$, $D(b(t))_i$ denotes the estimation of the reward of the i-th arm.

Using a DNN to predict the reward of each arm allows our algorithm to learn a nonlinear representation of the context. This representation is later used for exploration by performing linear TS on top of the last hidden layer activations. We denote the activations of the last hidden layer of $D$ applied to this context as $\phi(t) = \text{LastLayerActivations}(D(b(t)))$, where $\phi(t) \in \mathbb{R}^g$. The context $b(t)$ represents raw measurements that can be high dimensional (e.g., image or text), where the size of $\phi(t)$ is a design parameter that we choose to be smaller ($g < d$). This makes contextual bandit algorithms practical for such data sets. Moreover, $\phi(t)$ can potentially be linearly realizable (even if $b(t)$ is not) since a DNN is a global function approximator (Barron, 1993) and the last layer is linear.

**1.1 Training.** Every $L$ iterations, we train $D$ for $P$ mini-batches. Training is performed by sampling experience tuples $\{b(\tau), a(\tau), r_{a(\tau)}(\tau)\}$ from the replay buffer $E$ (details below) and minimizing the mean squared error (MSE),

$$||D(b(\tau))_{a(\tau)} - r_{a(\tau)}(\tau)||_2^2, \tag{3}$$

where $r_{a(\tau)}$ is the reward that was received at time $\tau$ after playing arm $a(\tau)$ and observing context $b(\tau)$ (similar to Riquelme et al. (2018)). Notice that only the output of arm $a(\tau)$ is differentiated.

We **emphasize** that the DNN, including the last layer, are trained end-to-end to minimize Eq. (3).

---

**Algorithm 2** Limited Memory Neural-linear TS

---

Set $\forall i \in [1,..,N] : \Phi_i^0 = I_d, \hat{\mu}_i = \mu_i^0 = 0_d, \Phi_i = 0_{dxd}, f_i = 0_d, a_i = a_i^0, b_i = b_i^0$
Initialize Replay Buffer $E$, and DNN $D$
Define $\phi(t) \leftarrow \text{LastLayerActivations}(D(b(t)))$
**for** $t = 1, 2, \ldots,$ **do**
    **Observe** $b(t)$, evaluate $\phi(t)$
    **Posterior sampling:** $\forall i \in [1,..,N]$, sample:
      $\tilde{\nu}_i(t) \sim \text{Inv-Gamma}\,(a_i(t), b_i(t))$
      $\tilde{\mu}_i(t) \sim N\left(\hat{\mu}_i, \tilde{\nu}_i(t)^2(\Phi_i^0 + \Phi_i)^{-1}\right)$
    **Play** arm $a(t) := \arg\max_i \phi(t)^T \tilde{\mu}_i(t)$
    **Observe** reward $r_t$
    **Store** $\{b(t), a(t), r_t\}$ in $E$
    **if** $E$ is full **then**
      Remove the first tuple in $E$ with $a = a(t)$ (round robin)
    **end if**
    **Bayesian linear regression update:**
    $\Phi_{a(t)} = \Phi_{a(t)} + \phi(t)\phi(t)^T, f_{a(t)} = f_{a(t)} + \phi(t)^T r_t$
    $\hat{\mu}_{a(t)} = (\Phi_{a(t)}^0 + \Phi_{a(t)})^{-1}\left(\Phi_{a(t)}^0 \mu_{a(t)}^0 + f_{a(t)}\right)$
    $a_{a(t)} = a_{a(t)} + \frac{1}{2}$
    $R_{a(t)}^2 = R_{a(t)}^2 + r_t^2$
    $b_{a(t)} = b_{a(t)}^0 + \frac{1}{2}\left(R_{a(t)}^2 + (\mu_{a(t)}^0)^{\mathrm{T}} B_{a(t)}^0 \mu_{a(t)}^0 - \hat{\mu}_{a(t)}^{\mathrm{T}} B_{a(t)} \hat{\mu}_{a(t)}\right)$
    **if** $(t \bmod L) = 0$ **then**
      **for** $\forall i \in [1,..,N]$ **do**
        Evaluate old features on the replay buffer: $E_{\phi^{old}}^i$
      **end for**
      **Train** DNN for $P$ steps
      **Compute priors** for new features:
      **for** $\forall i \in [1,..,N]$ **do**
        Evaluate new features on the replay buffer: $E_\phi^i$
        Solve for $\Phi_i^0$ using Eq. (6) with $E_\phi^i, E_{\phi^{old}}^i, \Phi_i^{old}$
        Set $\mu_i^0 \leftarrow \text{LastLayerWeights}(D)_i$
        $\Phi_i = \sum_{j=1}^{n_i} \phi_j^i(\phi_j^i)^T, f_i = \sum_{j=1}^{n_i} (\phi_j^i)^T r_j$.
      **end for**
    **end if**
**end for**

---

**2. Exploration.** Since our algorithm is performing training in phases (every $L$ steps), exploration is performed using a fixed representation $\phi$ ($D$ has fixed weights between training phases). At each time step $t$, the agent observes a raw context $b(t)$ and uses the DNN $D$ to produces a feature vector $\phi(t)$. The features $\phi(t)$ are used to perform linear TS, similar to Algorithm 1, but with two key differences.

First, we introduce a likelihood matching mechanism that accounts for changes in representation (see 4. below for more details). Second, we follow the Bayesian linear regression equations, as suggested in (Riquelme et al., 2018), and perform TS while updating the posterior both for $\mu$, the mean of the estimate, and $\nu$, its variance.

This is done in the following manner. We begin by sampling a weight vector $\tilde{\mu}_i$ for each arm $i \in 1..N$, from the posterior by following two steps. First, the variance $\tilde{\nu}_i^2$ is sampled from Inv-Gamma $(a_i, b_i)$. Then, the weight vector $\tilde{\mu}_i$ is sampled, from $N\left(\hat{\mu}_i, \tilde{\nu}_i^2(\Phi_i^0 + \Phi_i)^{-1}\right)$. Once we sampled a weight vector for each arm, we choose to play arm $a(t) = \arg\max_i \phi(t)^T \tilde{\mu}_i$, and observe reward $r_{a(t)}(t)$. This is followed by a posterior update step, based on Eq. (2):

$$\Phi_{a(t)} = \Phi_{a(t)} + \phi(t)\phi(t)^T, \quad f_{a(t)} = f_{a(t)} + \phi(t)^T r_t, \quad R_{a(t)}^2 = R_{a(t)}^2 + r_{a(t)}(t)^2 \quad (4)$$

$$\hat{\mu}_{a(t)} = (\Phi_{a(t)} + \Phi_{a(t)}^0)^{-1}(\Phi_{a(t)}^0 \mu_{a(t)}^0 + f_{a(t)}), \quad a_{a(t)} = a_{a(t)} + \frac{1}{2},$$

$$b_{a(t)} = b_{0,a(t)} + \frac{1}{2}\Big(R_{a(t)}^2 + \mu_{0,a(t)}^T \Phi_{0,a(t)} \mu_{0,a(t)} - \hat{\mu}_{a(t)}(t)^T \Phi_{a(t)}(t)\hat{\mu}_{a(t)}(t)\Big).$$

The exploration mechanism is responsible for choosing actions; it **does not change** the weights of the DNN.

**3. Memory buffer.** After an action $a(t)$ is played at time $t$, we store the experience tuple $\{b(t), a(t), r_{a(t)}(t)\}$ in a finite memory buffer of size $n$ that we denote by $E$. Once $E$ is full, we remove tuples from $E$ in a round robin manner, i.e., we remove the first tuple in $E$ with $a = a(t)$.

**4. Likelihood matching.** Before each learning phase, we evaluate the features of $D$ on the replay buffer. Let $E_i$ be a subset of memory tuples in $E$ at which arm $i$ was played, and let $n_i$ be its size. We denote by $E_{\phi^{old}}^i \in \mathbb{R}^{n_i \times g}$ a matrix whose rows are feature vectors that were played by arm $i$. After a learning phase is complete, we evaluate the new activations on the same replay buffer and denote the equivalent set by $E_\phi^i \in \mathbb{R}^{n_i \times g}$.

Our approach is to summarize the knowledge that the algorithm has gained from exploring with the features $\phi^{old}$ into priors on the new features $\Phi_i^0, \mu_i^0$. Once these priors are computed, we restart the linear TS algorithm using the data that is currently available in the replay buffer. For each arm $i$, let $\phi_j^i = (E_\phi^i)_j$ be the j-th row in $E_\phi^i$ and let $r_j$ be the corresponding reward, we set $\Phi_i = \sum_{j=1}^{n_i} \phi_j^i (\phi_j^i)^T, f_i = \sum_{j=1}^{n_i} (\phi_j^i)^T r_j$.

We now explain how we compute $\Phi_i^0, \mu_i^0$. Recall that under the realizability assumption we have that $\mathbb{E}[r_i(t)|\phi(t)] = \phi(t)^T \mu_i = \phi^{old}(t)^T \mu_i^{old} = \mathbb{E}[r_i(t)|\psi(t)]$. Thus, the likelihood of the reward is invariant to the choice of representation, i.e. $N(\phi(t)^T \mu_i, \nu^2) \sim N(\phi^{old}(t)^T \mu_i^{old}, \nu^2)$. For all $i$, define the estimator of the reward as $\theta_i(t) = \phi(t)^T \tilde{\mu}_i(t)$, and its standard deviation $s_{t,i} = \sqrt{\phi(t)^T \Phi_i(t)^{-1}\phi(t)}$ (see (Agrawal & Goyal, 2013) for derivation). By definition of $\tilde{\mu}_i(t)$, marginal distribution of each $\theta_i(t)$ is Gaussian with mean $\phi_i(t)^T \hat{\mu}_i(t)$ and standard deviation $\nu_i s_{t,i}$. The goal is to match the likelihood of the reward estimation $\theta_i(t)$ given the new features to be the same as with the old features.

**4.1 Approximation of the mean $\mu_i^0$:** Recall that the realizability assumption implies a linear connection between $\mu_i^{old}, \mu_i^0$, i.e., $(\mu_i^{old})^T \phi^{old} = (\mu_i^0)^T \phi$, thus, we can solve a linear set of equations and get a linear mapping from $\mu_i^0$ to $\mu_i^{old}$:

$$(\mu_i^0)^T = (\mu_i^{old})^T E_{\phi^{old}}^i (E_\phi^i)^{-1}. \quad (5)$$

In addition to the realizability assumption, for Eq. (5) to hold the matrix $E_\phi^i$ must be invertible. In practice, we found that a different solution that is based on using the DNN weights performed better. Recall that the DNN is trained to minimize the MSE (Eq. (3)). Thus, given the new features $\phi$, the weights of the last layer of the DNN make a good prior for $\mu_i^0$. This approach was shown empirically to make a good approximation (Levine et al., 2017), as the DNN was optimized online by observing all the data (and is therefore not limited to the current replay buffer).

**4.2 Approximation of the variance $s_{j,i}$:.** For each arm $i$, our algorithm receives as input the sets of new and old features $E_\phi^i, E_{\phi^{old}}^i$; denote the elements in these sets by $\{\phi_j^{old}, \phi_j\}_{j=1}^{n_i}$. In addition, the algorithm receives the correlation matrix $\Phi_i^{old}$. Notice that due the nature of our algorithm, $\Phi_i^{old}$ holds

information on contexts that are not available in the replay buffer. The goal is to find a correlation matrix, $\Phi_i^0$, for the new features that will hold the same information on past context as $\Phi_i^{old}$. I.e., we want to find $\Phi_i^0$ such that $\forall i \in [1..N], j \in [1..n_i]$ $s_{j,i}^2 \doteq (\phi_j^{old})^T (\Phi_i^{old})^{-1} \phi_j^{old} = \phi_j^T (\Phi_i^0)^{-1} \phi_j$.

Using the cyclic property of the trace, this is equivalent to finding $\Phi_i^0$, s.t. $\forall j \in [1, .., n_i], s_{j,i}^2 =$ Trace $\left( (\Phi_i^0)^{-1} \phi_j \phi_j^T \right)$. Next, we define $X_i$ to be a vector of size $n_i$ in the vector space of $d \times d$ symetric matrices, with its j-th element $X_{j,i}$ to be the matrix $\phi_j \phi_j^T$. Notice that $(\Phi_i^0)^{-1}$ is constrained to be semi positive definite (being a correlation matrix), thus, the solution can be found by solving an SDP (Eq. (6)). Note that Trace$(X_{j,i}^T (\Phi_i^0)^{-1})$ is an inner product over the vector space of symmetric matrices, known as the Frobenius inner product. Thus, the optimization problem is equivalent to a linear regression problem in the vector space of PSD matrices. In practice, we use cvxpy (Diamond & Boyd, 2016) to solve for all actions $i \in [1..N]$ :

$$\underset{(\Phi_i^0)^{-1}}{\text{minimize}} \sum_{j=1}^{n_i} ||\text{Trace}(X_{j,i}^T (\Phi_i^0)^{-1}) - s_{j,i}||^2 \quad \text{subject to} \quad (\Phi_i^0)^{-1} \succeq 0. \tag{6}$$

**Discussion.** The correctness of our algorithm follows from the proof of (Agrawal & Goyal, 2013). To see this, recall that we match the moments of the reward estimate $\theta_i(t)$ after every time that the representation changes. Assuming that we solve Eq. (5) and Eq. (6) precisely, then the reward estimation given the new features have precisely the same moments and distribution as with the old features. Since the distribution of the estimate did not change, its concentration and anti-concentration bounds do not change, and the proof in (Agrawal & Goyal, 2013) can be followed.

The problem is, that in general, we cannot guarantee to solve Eq. (5) and Eq. (6) exactly. We will soon show that under the realizability assumption, in addition to an invertibility assumption, it is possible to choose an analytical solution for the priors $\mu_0, \Phi^0$ that guarantees an exact solution. However, these conditions may be too strong and not realistic. We describe this scenario to highlight the existence of a scenario (and conditions) in which our algorithm is optimal; we hope to relax them in future work. We emphasize here that if these conditions do not hold, then our algorithm is only an approximation, without theoretical guarantees. In the next section we justify using our algorithm through thorough experimentation.

For simplicity, we consider a single arm. Assume that $m$ past observations, which we denote by $E_{\phi^{old}}^m$, were used to learn estimators $(\hat{\mu}_m)^{old}, \Phi_m^{old}$ using BLR (Eq. (2)). Due to the limited memory, some of these measurements are not available in the replay buffer, and all of the information regarding them is summarized in $(\hat{\mu}_m)^{old}, \Phi_m^{old}$. In addition, we are given a replay buffer of size $n$, that is used to produce (before and after the training) new and old feature matrices $E_{\phi^{old}}^n, E_\phi^n$. We also denote by $R_n$ the reward vector (using data from the replay buffer) and by $R_m$ the reward vector that corresponds to features $E_{\phi^{old}}^m$ which is not available in the replay buffer. Recall that the realizeability assumption implies that the features $\phi$ and $\phi^{old}$ are linear mappings of the raw context $b$, i.e., $\phi = A_\phi b, \phi^{old} = A_{\phi^{old}} b$. Under the assumption that all the relevant matrices are invertible, we use Eq. (5) to find a prior for $\mu_0$, i.e., we set $\mu_0^T = (\hat{\mu}_m^{old})^T E_{\phi^{old}}^n (E_\phi^n)^{-1}$. In addition, for the covariance matrix, we set $(\Phi^0)^{-1} = (E_\phi^n)^{-1} E_{\phi^{old}}^n (\Phi_m^{old})^{-1} (E_{\phi^{old}}^n)^T ((E_\phi^n)^T)^{-1}$, which is a solution to Eq. (6).

In addition, we get that if the relevant matrices are invertibele, then $\Phi^0 = \Phi_m$, and that $\Phi^0 \mu_0 = E_\phi^m R_m$ (see the supplementary for derivation). Plugging these estimates as priors in the Bayesian linear regression equation we get the following solution for $\hat{\mu}$ :

$$\hat{\mu} = (\Phi_n + \Phi^0)^{-1} (\Phi^0 \mu_0 + E_\phi^n R_n) = (\Phi_n + \Phi_m)^{-1} (E_\phi^m R_m + E_\phi^n R_n),$$

i.e., we got the linear regression solution for $\mu$ as if we were able to evaluate the new features $\phi$ on the entire data, while having a finite memory buffer and changing features!

## 3.1 COMPUTATIONAL COMPLEXITY

**Solving the SDP.** Recall that the dimension of the last layer is $g < d$ where $d$ is the dimension of the raw features, and the size of the buffer is $n$. Following this notation, when solving the SDP, we optimize over matrices in $\mathbb{R}^{g \times g}$ that are subject to $n$ equality constraints. We refer the reader to Vandenberghe & Boyd (1996) for an excellent survey on the complexity of solving SDPs. Here, we will refer to interior-point methods. The number of iterations required to solve an SDP to a given accuracy grows with problem size as $O(g^{0.5})$. Each iteration involves solving a least-squares

problem of dimension $g^2$. If, for example, this least-squares problem is solved with a projected gradient descent method, then the time complexity for finding an $\epsilon-$optimal solution is $g^2/\epsilon$, and the computational complexity of each gradient step is $ng^2$ (matrix-vector multiplications). Vandenberghe & Boyd experimented with solving SDPs of different sizes and observed that it takes almost the same amount of iterations to solve them. In addition, they found that SDP algorithms converge much faster than the worst-case theoretical bounds.

**In our case,** $g$, the size of the last layer, was fixed to be 50 in all the experiments. Thus, although the dimension of the raw features $d$ varies in size across data sets, the complexity of the solving the SDP is fixed. The dependence of the computational complexity on the buffer size $n$ is at most linear (CVXPY exploits sparsity structure of the matrix to enhance computations); we didn't encounter a significant changes in computation time when changing the buffer size in the range of $200 - 2000$. It took us $10 - 30$ seconds on a standard "MacBook Pro" to solve a single SDP.

**Dependence on** $T$**.** The full memory approach results in computational complexity of $O(T^2)$ and memory complexity of $O(T)$ where $T$ is the number of contexts seen by the algorithm. This is because it is estimating the TS posterior using the complete data every time the representation changes. On the other hand, the limited memory approach uses only the memory buffer to estimate the posterior but additionally solves an SDP. This gives a memory complexity of $O(1)$ and computational complexity of $O(T)$.

**Dependence on** $A$**.** The computational complexity is linear in the number of actions (we solve an SDP for each action). There is a large variety of problems where this is not an issue (as in our experiments). However, if the problem of interest has many discrete actions, our approach may not be useful.

To summarize, our method is more efficient than the full memory baseline in problems with big data (large $T$). Nevertheless, our method requires to solve an SDP (every $L$ iterations), which is computationally prohibitive in general. We deal with this issue by restricting the size of the last layer to be small ($g = 50$), for which solving the SDP is reasonable.

## 4 EXPERIMENTS

We begin this section by testing the resilience of our method to catastrophic forgetting. We present an ablative analysis of our approach and show that the prior on the covariance is crucial. Then, we present results for using MLPs on ten real-world data sets, including a high dimensional natural language data on a task of sentiment analysis (all of these data sets are publicly available through the UCI Machine Learning Repository). Additionally, in the supplementary material, we use synthetic data to test and visualize the ability of our algorithm to learn nonlinear representations during exploration. In all the experiments we used the same hyperparameters (as in (Riquelme et al., 2018)) for the model, and the same network architecture (an MLP with a single hidden layer of size 50). The only exception is with the text CNN (details below). The size of the memory buffer is set to be 100 per action.

### 4.1 CATASTROPHIC FORGETTING

We use the Shuttle Statlog data set (Newman et al., 2008), a real world, nonlinear data set. Each context is composed of 9 features of a space shuttle flight, and the goal is to predict the state of the radiator of the shuttle (the reward). There are $k = 7$ possible actions, and if the agent selects the right action, then reward 1 is generated. Otherwise, the agent obtains no reward ($r = 0$).

We experimented with the following algorithms: (1) Linear TS (Agrawal & Goyal, 2013, Algorithm 1) using the raw context as a feature, with an additional uncertainty in the variance (Riquelme et al., 2018). (2) Neural-Linear TS (Riquelme et al., 2018). (3) Our neural-linear TS algorithm with limited memory. (4) An ablative version of (3) that calculates the prior only for the mean, similar to (Levine et al., 2017).

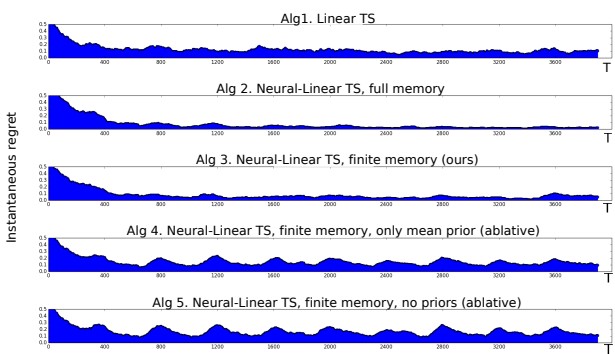

Figure 2: Catastrophic forgetting

(5) An ablative version of (3) that does not use prior calculations. Algorithms 3-5 make an ablative analysis for the limited memory neural-linear approach. As we will see, adding each one of the priors improves learning and exploration.

Fig. 2 shows the performance of each of the algorithms in this setup. We let each algorithm run for 4000 steps (contexts) and average each algorithm over 10 runs. The x-axis corresponds to the number of contexts seen so far, while the y-axis measures the instantaneous regret. All the neural-linear methods retrained the DNN every $L = 400$ steps for $P = 800$ mini-batches.

First, we can see that the neural linear method (2nd row) outperforms the linear one (1st row), suggesting that this data set in nonlinear. We can also see that our approach to computing the priors allows the limited memory algorithm (3rd row) to perform almost as good as the neural linear algorithm without memory constraints (2nd row).

In the last two rows we can see a version of the limited memory neural linear algorithm that does not calculate the prior for the covariance matrix (4th row), and a version that does not compute priors at all (5th row). Both of these algorithms suffer from "catastrophic forgetting" due to limited memory. Intuitively, the covariance matrix holds information regarding the number of contexts that were seen by the agent and are used by the algorithm for exploration. When no such prior is available, the agent explores sub-optimal arms from scratch every time the features are modified (every $L = 400$ steps, marked by the x-ticks on the graph). Indeed, we observe "peaks" in the regret curve for these algorithms (rows 4&5); this is significantly reduced when we compute the prior on the covariance matrix (3rd row), making the limited memory neural-linear bandit resilient to catastrophic forgetting.

## 4.2 REAL WORLD DATA

We evaluate our approach on several (10) real-world data sets; for each data set, we present the cumulative reward achieved by the algorithms, detailed above, averaged over 50 runs. Each run was performed for 5000 steps.

**Linear vs. nonlinear data sets:** The results are divided into two groups, linear and nonlinear data sets. The separation was performed post hoc, based on the results achieved by the full memory methods, i.e., the first group consists of five data sets on which Linear TS (Algorithm 1) outperformed Neural-Linear TS (Algorithm 2), and vice versa. We observed that most of the linear datasets consisted of a small number of features that were mostly categorical (e.g., the mushroom data set has 22 categorical features that become 117 binary features). The DNN based methods performed better when the features were dense and high dimensional.

| Name | d | A | Full memory | | Limited memory, Neural-Linear | | |
|---|---|---|---|---|---|---|---|
| | | | Linear | Neural-Linear | Both Priors | $\mu$ Prior | No Prior |
| Linear Data Sets | | | | | | | |
| Mushroom | 117 | 2 | $11022 \pm 774$ | $10880 \pm 853$ | $10923 \pm 839$ | $9442 \pm 1351$ | $7613 \pm 1670$ |
| Financial | 21 | 8 | $4588 \pm 587$ | $4389 \pm 584$ | $4597 \pm 597$ | $4311 \pm 598$ | $4225 \pm 594$ |
| Jester | 32 | 8 | $14080 \pm 2240$ | $12819 \pm 2135$ | $9624 \pm 2186$ | $10996 \pm 2013$ | $11114 \pm 2050$ |
| Adult | 88 | 2 | $4066.1 \pm 11.03$ | $4010.0 \pm 22.19$ | $3943.0 \pm 54.29$ | $3839.5 \pm 17.63$ | $3608.2 \pm 34.94$ |
| Covertype | 54 | 7 | $3054 \pm 557$ | $2898 \pm 545$ | $2828 \pm 593$ | $2347 \pm 615$ | $2334 \pm 603$ |
| Nonlinear Data Sets | | | | | | | |
| Census | 377 | 9 | $1791.5 \pm 39.47$ | $2135.5 \pm 51.47$ | $2023.16 \pm 37.3$ | $1873 \pm 757$ | $1943.83 \pm 84.2$ |
| Statlog | 9 | 7 | $4483 \pm 353$ | $4781 \pm 274$ | $4825 \pm 305$ | $4681 \pm 285$ | $4623 \pm 276$ |
| Epileptic | 178 | 5 | $1202.9 \pm 34.68$ | $1706.9 \pm 41.26$ | $1716.8 \pm 60.44$ | $1572.9 \pm 48.66$ | $1411.0 \pm 33.43$ |
| Smartphones | 561 | 6 | $3085.8 \pm 24.64$ | $3643.5 \pm 64.89$ | $2660.4 \pm 84.72$ | $3064.5 \pm 55.06$ | $2851.6 \pm 58.77$ |
| Scania Trucks | 170 | 2 | $4691.8 \pm 7.23$ | $4784.7 \pm 6.05$ | $4742.0 \pm 33.0$ | $4698.0 \pm 13.06$ | $4470.4 \pm 37.9$ |

Table 1: Cumulative reward of TS algorithms on 10 real world data sets. The context dim $d$ and the size of the action space $A$ are reported for each data set. The mean result and standard deviation of each algorithm is reported for 50 runs.

**Linear data sets:** Since there is no apriori reason to believe that real world data sets should be linear, we were surprised that the linear method made a competitive baseline to DNNs. To investigate this further, we experimented with the best reported MLP architecture for the covertype data set (taken from Kaggle). Linear methods were reported (link) to achieve around 60% test accuracy. This number is consistent with our reported cumulative reward (3000 out 5000). Similarly, DNNs achieved around 60% accuracy, which indicates that the Covertype data set is indeed relatively linear. However, when we measure the cumulative reward, the deep methods take initial time to learn, which can explain the slightly worst score. One particular architecture (MLP with layers 54-500-800-7) was reported to achieve 68%; however, we didn't find this architecture to yield better cumulative reward.

Similarly, for the Adult data set, linear and deep classifiers were reported to achieve similar results (link) (around $84\%$), which is again equivalent to our cumulative reward of 4000 out of 5000. A specific DNN was reported to achieve $90\%$ test accuracy but did not yield improvement in cumulative reward. These observations can be explained by the different loss function that we optimize or by the partial observably of the bandit problem (bandit feedback). Alternatively, competitions tend to suffer from overfitting in model selection (see the "reusable holdout" paper for more details (Dwork et al., 2015)). Regret, on the other hand, is less prune to model overfitting, because the model is evaluated at each iteration, and because we shuffle the data at each run.

**Limited memory:** Looking at Table 1 we can see that on *eight out of ten data sets*, using the prior computations (Algorithm 3), improved the performance of the limited memory Neural-Linear algorithms. On four out of ten data sets (Mushroom, Financial, Statlog, Epileptic), Algorithm 3 even outperformed the unlimited Neural-Linear algorithm (Algorithm 2).

**Limited memory neural linear vs. linear:** as linear TS is an online algorithm it can store all the information on past experience using limited memory. Nevertheless, in four (out of five) of the nonlinear data sets the limited memory TS (Algorithm 3) outperformed Linear TS (Algorithm 1). Our findings suggest that when the data is indeed not linear, than neural-linear bandits beat the linear method, even if they must perform with limited memory. In this case, computing priors improve the performance and make the algorithm resilient to catastrophic forgetting.

### 4.3 SENTIMENT ANALYSIS FROM TEXT USING CNNS

We use the "Amazon Reviews: Unlocked Mobile Phones" data set, which contains reviews of unlocked mobile phones sold on "Amazon.com". The goal is to find out the rating (1 to 5 stars) of each review using only the text itself. We use our model with a Convolutional Neural Network (CNN) that is suited to NLP tasks (Kim, 2014; Zahavy et al., 2018b). Specifically, the architecture is a shallow word-level CNN that was demonstrated to provide state-of-the-art results on a variety of classification tasks by using word embeddings, while not being sensitive to hyperparameters (Zhang & Wallace, 2015). We use the architecture with its default hyper-parameters (Github) and standard pre-processing (e.g., we use random embeddings of size 128, and we trim and pad each sentence to a length of 60). The only modification we made was to add a linear layer of size 50 to make the size of the last hidden layer consistent with our previous experiments.

Since the input is in $\mathbb{R}^{7k}$ ($60 \times 128$), we did not include a linear baseline in these experiments as it is impractical to do linear algebra (e.g., calculate an inverse) in this dimension. Instead, we focused on comparing our final method with the full memory neural

| $\epsilon-$greedy | Neural-Linear | Neural-Linear Limited Memory |
|---|---|---|
| $2963.9 \pm 68.5$ | $3155.6 \pm 34.9$ | $3143.9 \pm 33.5$ |

Figure 3: Cumulative reward on Amazon review's

linear TS and both prior computations with an $\epsilon-$greedy baseline. We experimented with 10 values of $\epsilon$, $\epsilon \in [0.1, 0.2, ..., 1]$ and report the results for the value that performed the best (0.1). Looking at Fig. 3 we can see that the limited memory version performs almost as good as the full memory, and better than the $\epsilon-$greedy baseline.

## 5 DISCUSSION

We presented a neural-linear contextual bandit algorithm that is resilient to catastrophic forgetting and demonstrated its performance on several real-world data sets. Our algorithm showed comparable results to a previous method that stores all the data in a replay buffer. The method requires to solve an SDP, which is computationally prohibitive in general. Thus, we restricted the size of the last layer to be small, such that solving the SDP is feasible.

To design our algorithm, we assumed that all the representations that are produced by the DNN are realizable. In practice, the features that are learned in the first iterations are nearly realizable and further iterations improve them. We hope to relax these assumptions in future work.

Our algorithm presented excellent performance on multiple real-world data sets. Moreover, its performance did not deteriorate due to the changes in the representation and the limited memory. We believe that our findings make an important step towards solving problems where both exploration and representation learning play an important role.

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

## APPENDIX A    ADDITIONAL SIMULATIONS: NON LINEAR REPRESENTATION LEARNING ON A SYNTHETIC DATA SET

**Setup:** we adapted a synthetic data set, known as the "wheel bandit" (Riquelme et al., 2018), to investigate the exploration properties of bandit algorithms when the reward is a nonlinear function of the context. Specifically, contexts $x \in \mathbb{R}^2$ are sampled uniformly at random in the unit circle, and there are $k = 5$ possible actions.

One action , $a_5$, always offers reward $r_5 \sim N(\mu_5, \sigma)$, independently of the context. The reward of the other actions depend on the context and a parameter $\delta$, that defines a $\delta-$circle $\|x\| \leq \delta$.

For contexts that are outside the circle, actions $a_1, .., a_4$ are equally distributed and sub-optimal, with $r_i \sim N(\mu, \sigma)$ for $\mu < \mu_5, i \in [1..4]$.

For contexts that are inside a circle, the reward of each action depends on the respective quadrant. Each action achieves $r_i \sim N(\mu_i, \sigma)$, where $\mu_5 < \mu_i = \dot{\mu}$ in exactly one quadrant, and $\mu_i = \mu < \mu_5$ in all the other quadrants. For example, $\mu_1 = \dot{\mu}$ in the first quadrant $\{x : \|x\| \leq \delta, x_1, x_2 > 0\}$ and $\mu_1 = \mu$ elsewhere. We set $\mu = 0.1, \mu_5 = 0.2, \dot{\mu} = 0.4, \sigma = 0.1$. Note that the probability of a context randomly falling in the high-reward region is proportional to $\delta$. For lower values of $\delta$, observing high rewards for arms $a_1, .., a_4$ becomes more scarce, and the role of the nonlinear representation is less significant.

We train our model on $n = 4000$ contexts, where we optimize the network every $L = 200$ steps for $P = 400$ mini batches. The results can be seen in Table 2.

Not surprisingly, the neural-linear approaches, even with limited memory, achieved better reward than the linear method (Table 2) [2].

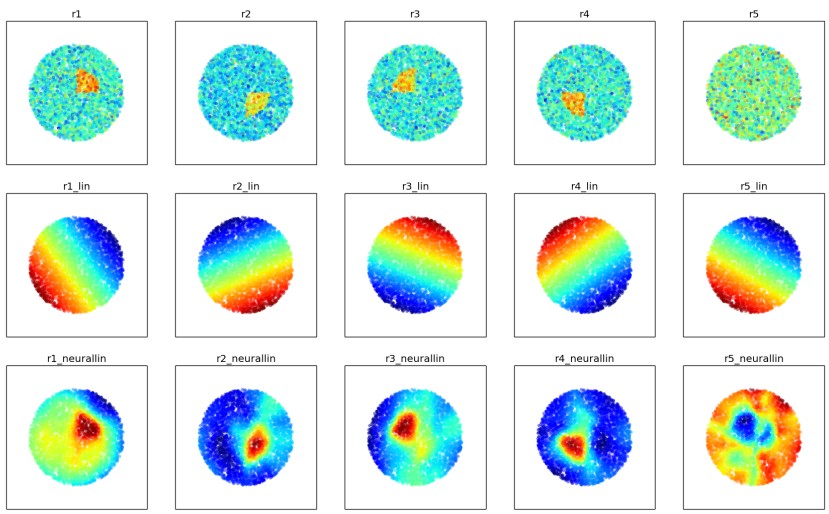

Figure 4: Representations learned on the wheel data set with $\delta = 0.5$. Reward samples (top), linear predictions (middle) and neural-linear predictions (bottom). Columns correspond to arms.

Fig. 4 presents the reward of each arm as a function of the context. In the top row, we can see empirical samples from the reward distribution. In the middle row, we see the predictions of the linear bandit. Since it is limited to linear predictions, the predictions become a function of the distance from the learned hyper-plane. This representation is not able to separate the data well, and also makes mistakes due to the distance from the hyperplane. For the neural linear method (bottom row), we can see that the DNN was able to learn good predictions successfully. Each of the first four arms learns to make high predictions in the relevant quadrant of the inner circle, while arm 5 makes higher predictions in the outer circle.

---

[2] We will provide a detailed comparison of the neural-linear algorithms and priors later in this section.

| | Linear | Neural-Linear Limited Memory |
|---|---|---|
| $\delta=0.5$ | $737.44 \pm 3.04$ | $899.72 \pm 12.79$ |
| $\delta=0.3$ | $735.37 \pm 2.58$ | $781.09 \pm 11.34$ |
| $\delta=0.1$ | $735.51 \pm 2.59$ | $751.75 \pm 3.6$ |

Table 2: Cumulative reward on the wheel bandit

## APPENDIX B   ANALYSIS

### B.1   DERIVATION OF AUXILIARY RESULTS FOR THE SANITY CHECK

The realizability assumption gives us a method to compute $\mu_0$ for the new features :

$$\mu_0^T = (\hat{\mu}_m^{old})^T E_{\phi^{old}}^n (E_\phi^n)^{-1} = \left( (E_{\phi^{old}}^m (E_{\phi^{old}}^m)^T)^{-1} (E_{\phi^{old}}^m)^T R_m \right) E_\phi^n (E_{\phi^{old}}^n)^{-1}. \tag{7}$$

Similarly, using the analytically solution to the SDP, we get

$$E_\phi^n (\Phi^0)^{-1} (E_\phi^n)^T = E_{\phi^{old}}^n (\Phi_m^{old})^{-1} (E_{\phi^{old}}^n)^T, \tag{8}$$

Using Eq. (7) and Eq. (8) and rearranging we get that

$$\begin{aligned}
\Phi^0 \mu_0 &= \\
&= E_\phi^n (E_{\phi^{old}}^n)^{-1} E_{\phi^{old}}^m R_m \\
&= (A_\phi b_n)(A_{\phi^{old}} b_n)^{-1} A_{\phi^{old}} b_m R_m \\
&= A_\phi b_m b_m^{-1} A_{\phi^{old}}^{-1} A_{\phi^{old}} b_m R_m \\
&= A_\phi b_m R_m = E_\phi^m R_m.
\end{aligned}$$

Similarly, for $\Phi^0$ we get that

$$\begin{aligned}
\Phi^0 &= E_\phi^n (E_{\phi^{old}}^n)^{-1} (\Phi_m^{old})((E_{\phi^{old}}^n)^T)^{-1} (E_\phi^n)^T \\
&= (A_\phi b_n)(A_{\phi^{old}} b_n)^{-1} (\Phi_m^{old})(b_n^{-1} A_{\phi^{old}}^{-1})^T (A_\phi b_n)^T \\
&= A_\phi A_{\phi^{old}}^{-1} A_\psi \sum_{i=1}^m b_i b_i^T A_{\phi^{old}}^T (A_{\phi^{old}}^T)^{-1} A_\phi^T \\
&= A_\phi \sum_{i=1}^m b_i b_i^T A_\phi^T = \Phi_m.
\end{aligned}$$

