# OpenReview forum: "Neural Linear Bandits: Overcoming Catastrophic Forgetting through Likelihood Matching"
_ICLR.cc/2020/Conference — Reject_

### Official Review · AnonReviewer3 · 2019-10-23
**Official Blind Review #3**

**Rating:** 3

**Review:**

Summary:

This work provides a memory-efficient nonlinear bandit algorithm based on deep neural networks. More specifically, the algorithm in this work only uses part of history information to save the memory usage. To overcome the catastrophic forgetting problem,  the authors provided novel covariance matrix approximation method. Experiment results also suggest that

Pros:

The writing of this paper is very well. It provides enough introduction of the background of nonlinear bandit problems. The experiment settings and results are convincible.

Cons:
- The core idea lacks solid theoretical supports. There is no regret bound result in this paper. The reason why I think the authors should add such theoretical proof is that it seems that the idea to construct new prior matrix instead of old one to avoid the catastrophic forgetting is not related to deep neural network at all. Thus, given existing regret analysis for Thompson sampling on linear bandit problems, the authors should also provide a simple analysis on linear case to show that the construction of prior matrix is indeed meaningful.

- The experiment part does not show the accuracy the SDP solve needs. As the authors mentioned in Discussion part, below equation 6, it is very crucial to decide the accuracy the SDP solver needs. I suggest the authors add more details about the SDP solver in the experiment part.


Minor comments:

- The authors used DNN to minimize equation 3. Have the authors tried  a regularized MSE instead of (3)? I think to add a regularizer can further improve the results.
- At page 13, below equation 8: why the first equality lacks?

**Experience Assessment:**

I have read many papers in this area.

**Review Assessment: Checking Correctness Of Derivations And Theory:**

I carefully checked the derivations and theory.

**Review Assessment: Checking Correctness Of Experiments:**

I assessed the sensibility of the experiments.

**Review Assessment: Thoroughness In Paper Reading:**

I read the paper at least twice and used my best judgement in assessing the paper.

---

> ### Author Response · Authors · 2019-11-07
> **Respond to reviewer 3**
>
> We agree with the reviewer's assessment that the core contribution of this paper is experimental and not theoretical.
>
> We would like to comment that the idea "to construct new prior matrix instead of old one to avoid the catastrophic forgetting" is related to deep neural networks since when the representation is being learned, the previous matrix is no longer relevant. Learning the representation using a neural net is the reason that the matrix has to be updated and is fundamental to representation learning. Nevertheless, we agree that the SDP solution technique itself is not "standard" to NNs.
>
> The sensitivity of the complete algorithm to the SDP solver - for simplicity, we did not tune the SDP solver and did not experiment with too many configurations. We have found the default setting of the CVXPY SDP solver to be sufficient. In some experiments that we performed we observed that is possible to solve the SDP with less accuracy (which can reduce the runtime of solving the SDP) but we did not investigate this direction further.
>
> Regularized MSE instead of (3) - this is a good idea but we did not explore it.
> Question: On page 13, below equation 8: why the first equality lacks?
> Answer: typo, we will fix that.

---

### Official Review · AnonReviewer2 · 2019-10-23
**Official Blind Review #2**

**Rating:** 3

**Review:**

This paper proposes a neural linear bandits algorithm that is resilient to catastrophic forgetting when using limited memory.

The proposed algorithm Alg. 2 is similar to Thompson sampling for linear contextual bandits, Alg. 1, but using the last layer activation vectors as a linear feature, and also a different way of updating noise parameter prior and posterior is used based on Bayesian linear regression Eq. (2).

Alg. 2 also works with limited memory of history data, therefore after every time, the memory is refreshed, likelihood matching is used to calculate new Phi to make the likelihood (mean and variance) of reward estimation the same as it for the old feature. For mean matching, minimizing MSE Eq. (3) is used and for variance, solving PSD problem Eq. (6) is used.

The complexity of this algorithm is analyzed. And experiments are conducted to show that the proposed method is resilient to catastrophic forgetting and can achieve good cumulative reward results.

The proposed method is reasonable and the results look promising. However, I found several weak points as follows.

1. As mentioned Bayesian linear regression Eq. (2) is used to update noise prior and posterior, but this update has no theoretical guarantees as mentioned. As an algorithm mainly works under bandit settings, this is kind of undesirable.

2. This algorithm works with neural network-based features, but it is in nature not scalable as shown in the complexity analysis (linear dependence on action number). The linear feature is just replaced by the last layer activation of NNs. From this perspective, the experimental results just justify again that the NN feature is somehow powerful, which is as expected.

3. The likelihood matching can deal with catastrophic forgetting with limited history memory, which looks good. But the fact that it actually works for linear feature (last layer activation) together with realization assumption weakens this contribution a lot. The authors find using Eq. (3) is better than the exact mean matching Eq. (5), and there is no explanation for this, which kind of shows the proposed likelihood matching probably is not a good way when using full NNs rather than just linear features (last layer). On the other hand, the SDP seems also can only work under linear feature settings, and is not promising to be generalized to fully update for NNs.

Overall, this is a reasonable paper. However, on the one hand, as an algorithm mainly works under bandit settings, it is a lack of theoretical support. On the other hand, the linear feature setting weakens the contribution of likelihood matching to deal with catastrophic forgetting with limited memory. There are some questions of the proposed mean matching, and the matching is not able to generalize.

**Experience Assessment:**

I have published one or two papers in this area.

**Review Assessment: Checking Correctness Of Derivations And Theory:**

I assessed the sensibility of the derivations and theory.

**Review Assessment: Checking Correctness Of Experiments:**

I assessed the sensibility of the experiments.

**Review Assessment: Thoroughness In Paper Reading:**

I read the paper at least twice and used my best judgement in assessing the paper.

---

> ### Author Response · Authors · 2019-11-07
> **Respond to reviewer 2**
>
> 1. Thank you for raising this point. We focused on Bayesian Linear Regression and not on the vanilla Gaussian TS as it is performing better empirically (as was observed in Riquelme et al). Nevertheless, when developing intuition and examining the correctness of our algorithm, we did focus on the Gaussian TS formulation (Agrawal and Goyal).
>
> 2. This is a good point. Our work indeed confirms that automatically learning to represent the data is essential to learning (and outperforms a linear baseline) in some problems, which is indeed expected. However, we believe that the main point in our paper is that representation learning can be performed alongside efficient exploration via the neural-linear mechanism. As we expressed in the paper, we are not familiar with other efficient mechanisms that combine representation learning with exploration.
>
> 3. Indeed, the main contribution of our work is experimental - we demonstrated that the approach works well on multiple datasets.

---

### Official Review · AnonReviewer1 · 2019-10-24
**Official Blind Review #1**

**Rating:** 3

**Review:**

This paper adapts Bayesian linear regression to the setting of a limited memory replay buffer. The idea is to calibrate the prior mean and variance when the neural representation of context is updated. Overall the paper is well written and explained clearly. Some experiments are provided to show that the proposed method is able to achieve a performance competitive to Bayesian linear regression with infinite memory.

The result of this paper is interesting. But I am not sure if the current experimental results are convincing enough to justify the significance of the proposed method.
1. The results in Section 4.2 seem to be following the setting in Riquelme 2018. These datasets are all in a supervised learning setting. It is a bit disappointing that the proposed method is not tested on RL datasets.
2. No other baseline is provided in the experiments for comparison.
    a. There are other methods in the literature to overcome catastrophic forgetting of neural networks, e.g regularizing the update of the network. How would that be compared to the proposed method?
    b. What about other methods, like [1]?
3. Most of the experiment details are missing. For example, how is the reward defined in section 4.3? What is the overhead in computation in practice, especially for the SDP?


Other comments:
1. Why would solving a SDP require only O(g^{0.5}) in section 3.1?
2. In the discussion in section 3, even if equation (5) and (6) can be exactly solved, how does the heavy tailed problem mentioned in section 2 been solved?


[1] Elmachtoub, Adam N., et al. "A practical method for solving contextual bandit problems using decision trees." arXiv preprint arXiv:1706.04687 (2017).

**Experience Assessment:**

I do not know much about this area.

**Review Assessment: Checking Correctness Of Derivations And Theory:**

I assessed the sensibility of the derivations and theory.

**Review Assessment: Checking Correctness Of Experiments:**

I assessed the sensibility of the experiments.

**Review Assessment: Thoroughness In Paper Reading:**

I read the paper at least twice and used my best judgement in assessing the paper.

---

> ### Author Response · Authors · 2019-11-07
> **Reply to reviewer 1**
>
> 1. The results in Section 4.2 are in the contextual bandit setup, similar to Riquelme et al. This is indeed similar to a supervised learning setup but incorporates online decision making, as in the contextual bandit formulation. We would like to comment that in addition to the datasets in Riquelme et al we incorporated three more data sets: epileptic, Scania, and smartphones, which we found to be more challenging. In addition, in section 4.3 we added a sentiment analysis from text using CNNs data set which was also not included in Riquelme et al. Overall, these datasets are quite diverse, and we are therefore surprised with the reviewer's disappointment. Finally, we would like to comment that we made an independent and separate submission to ICLR that is focused on experiments in the RL setup (following the RLSVI algorithm). If merging those results will convince the reviewer to increase his score, we will be happy to do so.
>
> 2. The reviewer is correct that there are other methods to overcome the catastrophic forgetting of neural networks, however, these methods are designed to deal with the consequences of forgetting on inference and not on exploration. There is no direct connection between the problems and we chose to focus on the latter. Therefore, there is no reasonable baseline to implement. We did compare our method with the "infinite memory" baseline of Riquelme et al and performed an extensive ablation study of our method.
>
> 3. The reward in Section 4.3 is 1 if the correct arm (sentiment) is chosen and 0 otherwise. We provided details regarding the computational efforts in solving the SDP. Roughly it takes 10 seconds to solve a single SDP. The computational benefit of using our method is apparent in large data sets, where the infinite memory method of Riquelme et al is becoming computationally prohibitive.
>
> Additional notes:
> 1. Answering the reviewer's question is out of the scope of this rebuttal. In the paper, we provided a reference for a survey paper on the computational complexity of solving SDPs where a detailed solution for this question can be found.
> 2. Thank you for raising this point. In section 3, we refer to the standard TS algorithm (with a fixed sigma) which has proved regret bounds and not to the bayesian linear regression formulation that we presented in the paper. BLR is working better in practice and since the focus of our paper is empirical we chose to present the algorithm with it and not with the simple TS. We will make this point clearer in a future revision.

---

> > ### Comment · AnonReviewer1 · 2019-11-15
> > **After Rebuttal**
> >
> > I have read other reviews and the response. The authors' response helps clarify some of my questions. I am happy to increase the score to weak accept.
> >
> > That being said, given the contribution of this paper is mainly focusing on the 'finite memory' aspect, I believe it is necessary for the paper to show the computational efficiency of the proposed method, including comparison to the full memory method, SDP solver, memory sizes etc..

---

### Decision · Program_Chairs · 2019-12-19

**Decision:**

Reject

**Comment:**

Reviewers found the problem statement having merit, but found the solution not completely justifiable. Bandit algorithms often come with theoretical justification because the feedback is such that the algorithm could be performing horribly without giving any indication of performance loss. With neural networks this is obviously challenging given the lack of supervised learning guarantees, but reviewers remain skeptical and prefer not to speculate based on empirical results.